# High-Throughput Phenotyping of Clinical Text Using Large Language Models

Daniel B. Hier
*Dept. of Neurology & Rehabilitation*
*University of Illinois at Chicago*
Chicago, IL USA
dhier@uic.edu

S. Ilyas Munzir
*Dept. of Neurology & Rehabilitation*
*University of Illinois at Chicago*
Chicago, IL USA
smunz2@uic.edu

Anne Stahlfeld
*Dept. of Neurology & Rehabilitation*
*University of Illinois at Chicago*
Chicago, IL USA
astahl5@uic.edu

Tayo Obafemi-Ajayi
*Engineering Program*
*Missouri State University*
Springfield, MO USA
tayoobafemiajayi@missouristate.edu

Michael D. Carrithers
*Dept. of Neurology & Rehabilitation*
*University of Illinois at Chicago*
Chicago, IL USA
mcar1@uic.edu

*Abstract*—**High-throughput phenotyping automates the mapping of patient signs to standardized concepts, such as those in Human Phenotype Ontology (HPO), a process critical to precision medicine. We evaluated the automated phenotyping of clinical summaries from the Online Mendelian Inheritance in Man (OMIM) database using a large language model. Various APIs were used to automate text retrieval, sign identification, categorization, and normalization. GPT-4 outperformed GPT-3.5-Turbo in identifying, categorizing, and normalizing signs, achieving concordance with manual annotators comparable to concordance between manual annotators. While GPT-4 demonstrates high accuracy in sign identification and categorization, limitations remain in sign normalization, particularly in retrieving the correct HPO ID for a normalized term. Methods such as retrieval-augmented generation, changes in pre-training, and additional fine-tuning may help address these limitations. The combination of APIs with large language models presents a promising approach for high-throughput phenotyping of free text.**

*Index Terms*—**phenotype, large language model, natural language processing, high-throughput, OMIM, neurology, HPO, GPT-4.**

## I. Introduction

Manual phenotyping of electronic health records is laborious and time-consuming [1], [2]. Precision medicine has driven the need for high-throughput phenotyping methods capable of processing large volumes of unstructured medical data efficiently [3], [4]. However, automating this process remains a challenge due to the complexity of medical text and the volume of physician notes [5]–[7]. Traditional natural language processing (NLP) methods for identifying phenotypic signs in clinical text have evolved from rule-based and dictionary-based systems [8], [9], to machine learning models [10], [11], and more recently to deep learning methods such as recurrent neural networks and convolutional neural networks [12]–[14]. Despite these advances, limitations remain, including low levels of accuracy, the need for large amounts of manually annotated data to train models, and the inability to generalize models from one medical domain to another [2], [6], [7], [15].

The emergence of large language models (LLM) offers an opportunity to overcome some of these challenges, particularly for high-throughput phenotyping [16]–[18]. An LLM, such as GPT-4, is capable of understanding and generating human-like text across various domains due to pretraining on various data sources [19]. These models demonstrate strong zero-shot, one-shot, and few-shot learning abilities, allowing them to perform complex tasks such as extracting, categorizing, and normalizing clinical phenotypes without additional training [19]. Recent work [20], [21] has shown the potential of an LLM to automate the phenotyping process for large-scale electronic health records (EHR) and clinical ssummaries. An LLM can also derive phenotypes from other sources such as PubMed abstracts and clinical summaries [22].

Precision medicine relies on accurately computed patient phenotypes to guide treatment decisions and improve outcomes [23]. However, patient phenotypes recorded in EHRs are unstructured and require extraction, categorization, and normalization before they can be entered into precision medicine machine learning models. Human Phenotype Ontology (HPO) [1], [24], [25] is the most widely used standard to record phenotypic information, providing a structured vocabulary to describe the signs and symptoms of the disease. In the following, we refer to the signs and symptoms of the disease as *signs*. The Online Mendelian Inheritance in Man (OMIM) database organizes diseases into *phenotypic series*—a collection of diseases with similar clinical features but caused by mutations in different genes. For example, the dystonia phenotypic series includes DYT6, DYT11, and DYT25, all characterized by involuntary muscle contractions but with different underlying genetic causes [26], [27]. Automating the identification, categorization, and normalization of phenotypes from OMIM clinical summaries can serve as a useful surrogate for processing physician notes. This task benefits from the

TABLE I
PHENOTYPIC SERIES PROCESSED BY HIGH-THROUGHPUT
PHENOTYPING

| Phenotypic Series | PS MIM | Diseases |
|---|---|---|
| amyotrophic lateral sclerosis (ALS) | PS105400 | 35 |
| Charcot-Marie-Tooth disease (CMT) | PS118220 | 81 |
| dystonia | PS128100 | 37 |
| epilepsy generalized | PS600669 | 29 |
| episodic ataxia | PS160120 | 9 |
| familial febrile seizures | PS121210 | 17 |
| hereditary spastic paraparesis (HSP) | PS303350 | 83 |
| hyperekplexia | PS149400 | 4 |
| leukodystrophy, hypomyelinating | PS312080 | 27 |
| narcolepsy | PS161400 | 7 |
| nemaline myopathy | PS161800 | 13 |
| Parkinson | PS168600 | 33 |
| progressive supranuclear palsy | PS601104 | 3 |
| restless legs | PS102300 | 8 |
| spinocerebellar ataxia | PS105400 | 40 |
| striato nigral degeneration | PS609161 | 2 |

availability of OMIM text through an API and the absence of privacy regulations that govern patient data. Moreover, the phenotyping process in OMIM clinical summaries is similar to that required for physician notes [21].

In this study, we evaluated two large language models, specifically GPT-4 and GPT-3.5-Turbo, for high-throughput phenotyping of clinical text. By automating the process through APIs, we assess the capabilities of the models to identify, categorize, and normalize clinical signs. Furthermore, we visualize variability within a neurological disease phenotypic series using heat maps and dimension-reduced scatter plots, providing insights into the diversity of disease phenotypes.

## II. DATA

Neurological disease phenotypic data were retrieved from the OMIM database using the API (api.omim.org). For each disease within the OMIM database, the disease phenotypes are described in the *clinical synopsis* and the *clinical characteristics* sections. The *clinical synopsis* is a list of signs, symptoms, mode of inheritance, and age of onset, while the *clinical features* section summarizes published literature that underpins the phenotype of each disease. The OMIM API has separate calls for clinical features and clinical synopsis. Diseases in OMIM with similar phenotypes are grouped in a phenotypic series. OMIM currently has 582 phenotypic series, each with an identifier beginning with PS. We evaluated 16 phenotypic series that spanned across 405 neurogenetic diseases (Table I).

## III. METHODS

Figure 1 outlines the high-throughput phenotyping pipeline used for phenotype term extraction and normalization. Detailed parameters for the OMIM API and OpenAI API calls, along with the Python code and data files are available at the project's GitHub site (https://github.com/clslabMSU/highthroughput-phenotyping).

*Text Extraction and Preprocessing.* Given a list of diseases and MIM numbers for each phenotypic series, the pipeline extracts clinical summaries from the OMIM API. White spaces and tabs were converted to a single white space. Punctuation, including commas, hyphens, semicolons, single quotes, double quotes, forward slashes, and backslashes, were also standardized to a single white space. Periods were retained to identify sentence boundaries.

*Sign Identification.* The extracted text was passed to the OpenAI API with a structured prompt to identify neurological signs and symptoms (Box 1). This prompt format was designed to ensure consistency and clarity in extracting relevant signs while minimizing ambiguity.

*Sign Categorization.* The identified signs were categorized into 30 high-level categories (Box 2) using a subsequent OpenAI API call. The structured prompt was used to ensure that the signs were classified accurately into clinically relevant categories.

*Sign Normalization.* Signs were normalized by mapping them to the Human Phenotype Ontology (HPO) using two approaches. The first approach utilized spaCy (Explosion AI, Berlin) combined with Gensim BioWordVec embeddings. Vectors were generated for each sign and compared to HPO terms using cosine similarity, with the highest similarity assigned as the best match. The second approach involved GPT-4 or GPT-3.5-Turbo, where the models were tasked with mapping signs to HPO terms and IDs (Box 3).

---

**Box 1:** Prompt for Sign Identification

You are a neurologist analyzing a case summary from OMIM. Your input is text containing 'Clinical Features' and 'Description'. Extract relevant neurological symptoms (patient complaints) and signs (findings on examination). Here's how the output should look:

'Signs': ['symptom a', 'symptom b', 'symptom c']

---

**Box 2:** Prompt for Sign Categorization

You are a neurologist analyzing a list of signs. Classify each sign into one of these categories:

'Behavior,' 'Bowel and Bladder,' 'Cognitive,' 'Deformity,' 'Dysautonomia,' 'Dystonia,' 'Extraocular Movements,' 'Fatigue,' 'Gait,' 'Head Shape,' 'Hearing,' 'Hyperkinesia,' 'Hyperreflexia,' 'Hypertonia,' 'Hypokinesia,' 'Hyporeflexia,' 'Hypotonia,' 'Incoordination,' 'Muscle Atrophy,' 'Other Cranial Nerve,' 'Pain,' 'Seizure,' 'Sensory,' 'Skin,' 'Sleep,' 'Speech,' 'Tremor,' 'Unclassified,' 'Vision,' 'Weakness.'

Your output should be a JSON object with each category as a key and a list of signs in that category as items.

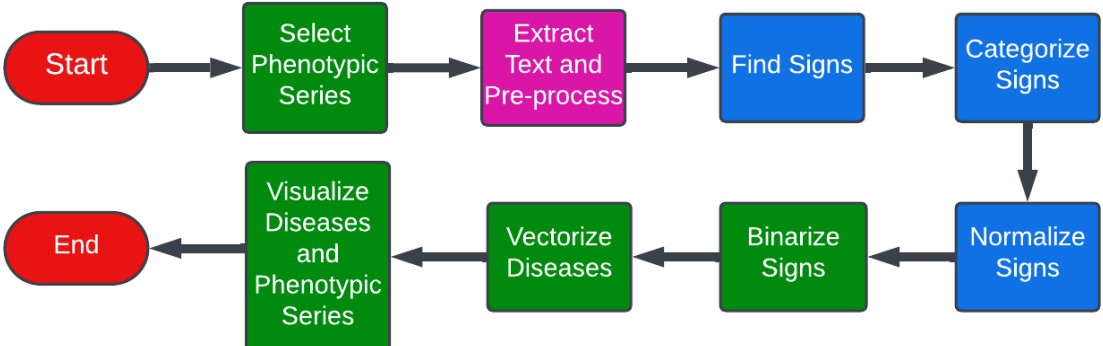

Fig. 1. **Pipeline for high-throughput phenotyping of clinical summaries from OMIM**. To support high-throughput, text retrieval, sign identification, sign categorization, and sign normalization is performed by an API.

**Box 3:** Prompt for Sign Normalization

You are a neurologist tasked with mapping each sign to a concept in the Human Phenotype Ontology (HPO). Your output should be a JSON object with each input sign as a key and two item values: the 'HPO Term' and the 'HPO ID.'
For example:

{'input': 'Apraxia oral,'
'HPO Term': 'Oromotor apraxia,'
'HPO ID': 'HP:0000687'}

If the input term cannot be mapped to HPO, return 'not-mappable' in the 'HPO Term' and 'HPO ID' fields.

*Category Binarization.* To facilitate analysis, the 30 phenotype categories were binarized as either '0' (no signs found in that category) or '1' (one or more signs present). This binarization simplifies downstream analysis by reducing the data to presence/absence values, enabling easier comparisons of phenotypic similarities across diseases.

*Disease Vectorization.* For each disease, a vector was constructed from the 30 binary phenotype categories. Each element of the vector represents the presence ('1') or absence ('0') of a phenotype in the corresponding category. Among the 405 diseases evaluated, 283 had adequate clinical summaries for high-throughput phenotyping, and their disease vectors were stored as a data frame.

*Visualization of Disease Heterogeneity within a Phenotypic Series.* Heatmaps were created for each phenotypic series to visualize the heterogeneity of phenotypic presentations. Each row in the heatmap represents a disease, and each column represents one of the 30 binary phenotype categories (**red** for 'present', **blue** for 'absent').

*Visualization of Distances between the Centroids of Phenotypic Series.* Principal Component Analysis (PCA) was used to reduce the 30 phenotype categories to two dimensions, allowing the visualization of distances between disease phenotypes in scatter plots (Fig. 7). Each centroid represents the phenotypic series' average position, visualized as an 'X' on the scatter plot. The relative proximity between phenotypic series centroids highlights similarities between the diseases within a phenotypic series.

*Performance Metrics.* Disease processing rates, sign identification rates, sign categorization rates, and sign normalization rates were calculated based on 405 diseases, 175,724 words, and 16 phenotypic series (Table I). Sign identification, categorization, and normalization were validated using a dataset of 40 diseases from the Dystonia, Parkinson, Hereditary Spastic Paraparesis, and Charcot-Marie-Tooth phenotypic series.

TABLE II
**PERFORMANCE METRICS**

| Model | GPT-3.5 Turbo | GPT-4 |
|---|---|---|
| Counts | | |
| Diseases | 405 | 405 |
| Usable Diseases | 207 | 283 |
| Signs Identified | 4,227 | 5,595 |
| Unique Signs Identified | 2,567 | 2,705 |
| Words Processed | 175,724 | 175,724 |
| Rates† | | |
| Disease Rate (sec/disease) | 14.2 | 16.4 |
| Identification Rate (sign/sec) | 5.7 | 4.2 |
| Categorization Rate (sign/sec) | 2.9 | 2.3 |
| Normalization Rate (sign/sec) | 9.3 | 9.3 |

† Performance times and rates are representative. They were obtained on Apple Mac Studio with an M2 ultra CPU running Mac OS 14.5.

## IV. RESULTS

We performed high-throughput neurological phenotyping on 405 disease variants from 16 OMIM phenotypic series (Table I). Sign identification, sign categorization, and sign normalization were performed by GPT-3.5-Turbo or GPT-4 in three sequential submissions to the OpenAI API. The running

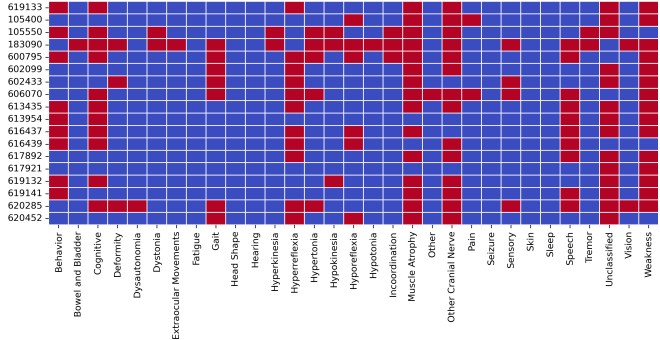

Fig. 2. **Heatmap for ALS phenotypic series with alphabetical category columns.** Diseases are along the y-axis. A unique MIM number identifies each disease. Compare to Fig. 3 with columns sorted by sign prevalence. Categories have been binarized so that 'red' indicates that the phenotype was present, and 'blue' indicates the phenotype was absent.

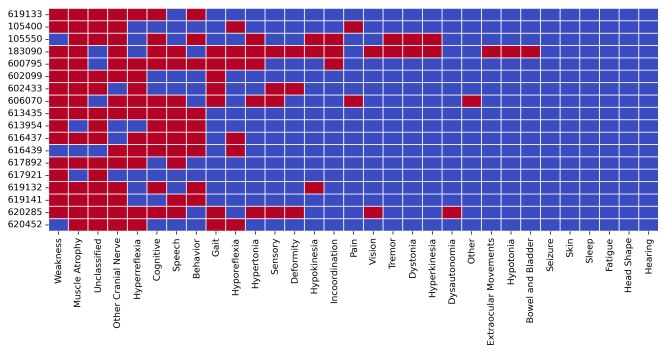

Fig. 3. **Heatmap for ALS phenotypic series with category columns sorted by sign prevalence** The most prevalent signs are weakness and muscle atrophy. Categories have been binarized so that 'red' indicates the phenotype was present and 'blue' indicates the phenotype was absent.

time per disease took 14.2s and 16.4s for GPT-3.5-Turbo and GPT-4 respectively. Although higher throughput might be possible with a faster CPU, more than 90% of the time expended was due to the four API calls.

The GPT-4 model outperformed the GPT-3.5-Turbo model on several performance metrics (Table II). GPT-4 produced usable data for 283 diseases, whereas GPT-3.5-Turbo produced usable data for 207 diseases. GPT-4 identified more signs (5,595 compared to 4,227) and more unique signs (2,705 compared to 2,567) than GPT-3.5-Turbo. The Jaccard Index, a stringent measure of concordance requiring exact matches between the large language models and the manual annotators, was higher for GPT-4 (0.31) than GPT-3.5-Turbo (0.16). A more relaxed measure of concordance, the maximum similarity index (based on cosine similarity from spaCy and BioWordVec embeddings from Gensim), showed high maximal mean similarities for signs compared to manual annotators (93.1 for GPT-3.5-Turbo and 94.2 for GPT-4). Weak matches (maximum similarity less than 0.80) were lower with GPT-4 than with GPT-3.5-Turbo. Compared to manual annotators, precision, recall, and F1 for sign identification were higher with GPT-4 than with GPT-3.5.

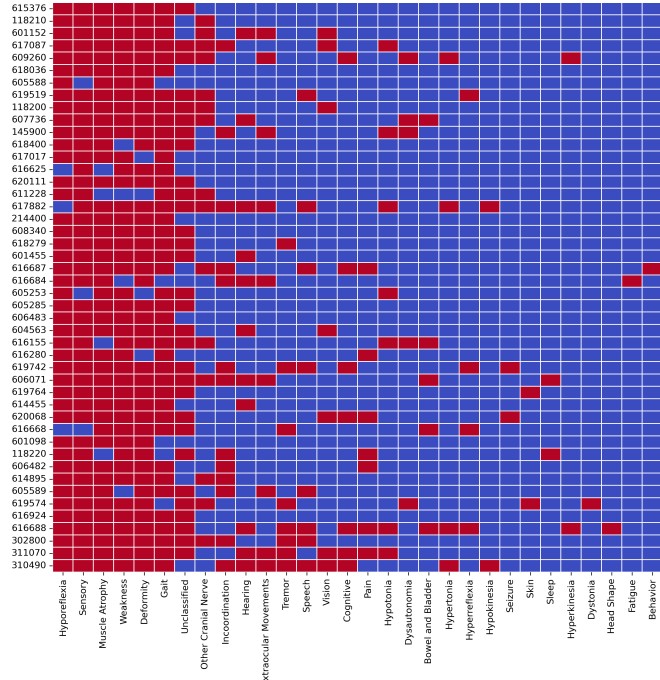

Fig. 4. **Heatmap for Charcot-Marie-Tooth phenotypic series.** The most prevalent signs are sensory symptoms, hyperreflexia, muscle atrophy, and weakness. MIM numbers for each disease in the phenotypic series are shown along the y-axis. Each row is a separate disease within the CMT phenotypic series and illustrates the diversity of phenotypic presentations of CMT within the phenotypic series. Categories have been binarized so that 'red' indicates the phenotype was present and 'blue' indicates the phenotype was absent.

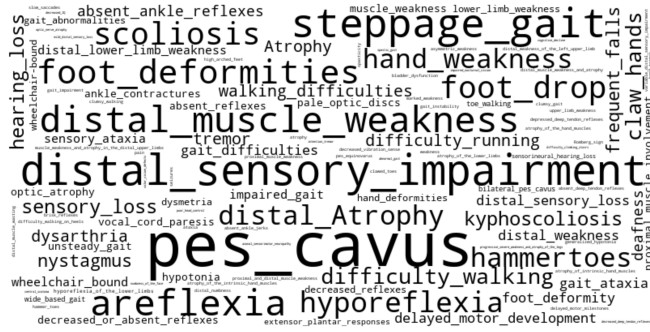

Fig. 5. **Word cloud for phenotypic terms for Charcot-Marie-Tooth disease phenotypic series.** 939 Terms were identified through GPT-4 API. Term size reflects relative frequency. Note that many similar terms include 'areflexia', 'hyperreflexia', and 'decreased or absent reflexes'. Compare to Fig. 5 after terms have been further categorized by GPT-4 API.

The OpenAI API interface assigned each sign to one of 30 high-level categories. A significant simplification of the feature space was achieved by categorization of signs, as illustrated by comparing the word clouds for CMT signs (Fig.5 with CMT categories (Fig. 5). The ability of GPT-3.5-Turbo and GPT-4 to correctly assign signs to high-level categories was manually checked by a neurology expert for signs in the disease validation set. The accuracy of the GPT-4 was higher than that of the GPT-3.5-Turbo on sign categorization (94.0%

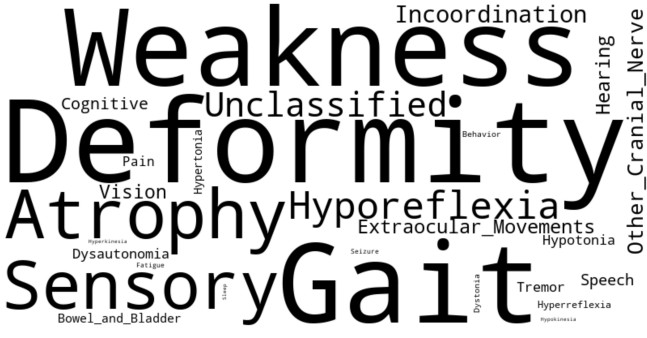

Fig. 6. **Word cloud for category frequencies for Charcot-Marie-Tooth (MCT) disease phenotypic series**. Phenotypic terms used to describe CMT diseases have been reduced to 30 categories. Word size in the word cloud reflects the size of each category. Compare to Fig. 4. The largest categories are Weakness, Deformity, and Gait.

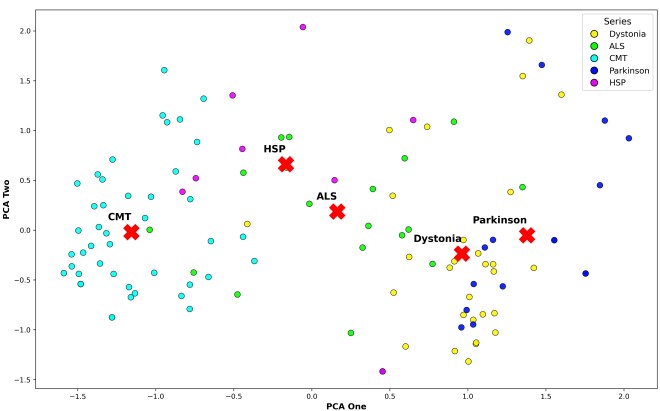

Fig. 7. **Centroids plotted by phenotypic series.** The feature space has been reduced from 30 high-level categories to 2 dimensions by PCA. Each round marker is a disease in one of the five plotted phenotypic series. The **X** indicates the centroids for each phenotypic series. The expected proximities between ALS and HSP (both with weakness and spasticity) and between Parkinson and Dystonia (both movement disorders) are visualized. Five of the 16 available phenotypic series centroids are shown. Creating centroid plots for any combinations of the phenotypic series in Table I is possible. Due to concerns about interpretability, we have limited centroids plots to no more than 5 phenotypic series per plot.

compared to 58.4%). Sign categorization allowed us to create heat maps for each phenotypic series in which rows were diseases and columns were phenotype categories, as illustrated by Figs. 2 to Fig. 4. Distances between phenotypic series centroids can be plotted using PCA for dimension reduction. Fig. 7 shows an example of five centroids in a series of phenotypes. Sign normalization (Table V ) was evaluated for the disease validation set. The SOTA NLP method performed best at 90.6% accuracy, followed by GPT-4 at 57.9% accuracy, and GPT-3.5-Turbo at 44.8% accuracy.

## V. DISCUSSION

We have developed a high-throughput pipeline that processes clinical text and identifies signs of disease. To support high-throughput, ease of use, and processing speed, the pipeline uses application programming interfaces (APIs) [28].

TABLE III
SIGN IDENTIFICATION METRICS

| Model | GPT-3.5-Turbo* | GPT-4* | Inter-Rater** |
|---|---|---|---|
| Signs Identified | 358 | 609 | 694 |
| Weak Matches (%) | 15.0 | 11.6 | 4.0 |
| Jaccard Index | 0.16 | 0.35 | 0.36 |
| Max Similarity Index | 93.1 | 94.2 | 96.7 |
| F1 | 0.52 | 0.66 | 0.60 |
| Precision | 0.61 | 0.66 | 0.96 |
| Recall | 0.45 | 0.65 | 0.44 |

*Concordance for sign identification between GPT-3.5-Turbo and GPT-4 with the two manual annotators for 40 diseases in the validation dataset.
**Inter-rater concordance for the manual annotators. Note that GPT-4 achieves a Jaccard Index similar to that between manual annotators.

TABLE IV
SIGN CATEGORIZATION METRICS

| Model | Accuracy | Precision | Recall |
|---|---|---|---|
| GPT-4 | 94.0 | 98.3 | 95.5 |
| GPT-3.5 Turbo | 58.4 | 78.4 | 95.2 |

Metrics based on manual review of sign categorization for the 40 diseases in the validation dataset.

We used an API to retrieve the summary text from OMIM and another API to allow GPT-4 to identify, categorize, and normalize signs. Clinical summaries from the OMIM database were utilized as our use case since the text is easily retrievable, rich with phenotypes, and not regulated as protected health information. However, these methods can be applied to text from other sources, including electronic health records, PubMed abstracts, full-text articles, and other clinical summaries.

Recognizing (identifying signs) and normalizing (mapping signs to an ontology) are challenging tasks for traditional NLP methods [2], [15], [29]–[32]. Progress has been made toward improving the recognition and normalization of medical concepts using transformers combined with specialized biomedical word embeddings [33], [34]. Large pre-trained language models provide a new approach to deep phenotyping (concept identification and normalization) that does not require additional training or a large corpus of manual annotations [22], [35]–[37]. Our pipeline for high-throughput phenotyping performed three phenotyping operations: sign identification, sign categorization, and sign normalization. In general, GPT-4 performed these operations with high accuracy and outperformed GPT-3.5-Turbo (Tables III, IV and V). Similarly, Groza et al. [22] evaluated GPT models for phenotype concept recognition using the ChatGPT interface. Their study demonstrated that GPT-4 outpaced the state-of-the-art methods in mention-level F1 scores of 0.7. Our work extends that of Groza et al. by demonstrating the utility of the GPT API to facilitate high-throughput phenotyping. In previous work, we have shown that GPT-4 can identify phenotypes in physician notes [20], [21], which is important for precision medicine [38], [39].

GPT-4 exhibited some weaknesses in sign normalization, achieving an accuracy of only 57.9%. This task has been noted by others as particularly challenging for GPT-4 [22]. In com-

TABLE V
SIGN NORMALIZATION METRICS

| Model | Accuracy | Precision | Recall |
|---|---|---|---|
| GPT-4 | 57.9 | 59.0 | 94.1 |
| GPT-3.5 Turbo | 44.8 | 49.8 | 52.9 |
| SOTA NLP | 90.6 | 90.8 | 99.8 |

Metrics based on manual review of the normaliza-
tion of signs of the 40 diseases in the validation
dataset. SOTA NLP is the spaCy cosine similarity
method with Gensim BioWordVec embeddings.

parison, a state-of-the-art NLP model (SOTA) that combined BioWordVec from Gensim with the spaCy NLP similarity method demonstrated significantly higher accuracy at 90.6%. Although GPT-4 excelled at identifying plausible HPO terms for each input term, it was notably less accurate in providing the correct HPO IDs. In some instances, it even produced implausible HPO IDs. This discrepancy likely stems from GPT-4's design, which relies heavily on pre-training to infer HPO IDs rather than employing a direct lookup capability. Currently, GPT-4 does not have an inherent mechanism to verify or retrieve accurate HPO IDs from a database. Shlyk et al. [40] have suggested remedying this limitation by adding retrieval augmented generation to assist in finding the correct HPO ID. Moreover, an inherent limitation of GPT models like GPT-4 is their non-deterministic nature. The choice of HPO ID for sign normalization can vary between different runs, even when the same input is provided [22]. This variability introduces inconsistencies that can be problematic in clinical applications where reliability is paramount.

We used GPT-4 to categorize the signs into 30 high-level categories. These high-level categories were chosen for their relevance to neurological phenotypes [41]. Although HPO has 28 high-level categories under *Phenotypic abnormality* [42], these categories are too broad to be useful in analyzing the phenotypes of neurological diseases. This categorization process significantly reduced the number of phenotypic terms needed to describe the diseases (compare Fig. 5 to Fig. 6). By assigning each phenotypic term to one of 30 high-level categories, we gained the ability to represent each disease in a phenotypic series as a row on a heatmap (Figs. 2 to 4). Heatmaps have also been used to visualize Orphadata disease phenotypes [41].

Once the phenotypic terms are acquired, a disease phenotype can be represented as a vector. Various methods are available to calculate the similarity between these disease vectors [43]–[48]. We used Principal Component Analysis (PCA) to reduce the dimensionality of these vectors to two dimensions (x and y), enabling us to visualize each disease as a marker on a scatter plot. To visualize the distances between the phenotypic series, we represent each series as a centroid of its component diseases. Although Fig. 7 is representative, these methods can be applied to display phenotypic distances between any combination of diseases or phenotypic series.

Large language models, including GPT-4, show promise for high-throughput phenotyping of clinical text, though some

issues identified in this work warrant further investigation. The level of accuracy required by an LLM for clinical decision-making remains uncertain [49]. It is important to recognize that human annotators do not always agree perfectly [50], and even expert physicians are susceptible to diagnostic errors [51]. There is a debate over whether health informatic tasks, such as phenotyping, are better suited to large general-purpose models or smaller, specially trained language models [52]. Concerns have been raised about the foundational weaknesses of large language models in healthcare, stemming from their limited training on EHR data [53]. Furthermore, an LLM struggles to process EHR data in tabular form (for example, the long tables of biochemical results) [54]. Groza et al. [22] have highlighted the stochastic nature of LLM outputs. If these models are to be used routinely in healthcare, issues of trust, privacy, equity, fairness, and confidentiality must be satisfactorily addressed [55], [56]. Furthermore, the problem of 'hallucinations' and 'confabulations' by LLMs remains unresolved [57]. The reduced accuracy of the LLM in retrieving the correct HPO ID (a normalization task) is a notable limitation (Table V).

This work has some limitations. While we tested GPT-3.5-Turbo and GPT-4, we did not compare their performance with other proprietary or open-source models. Future work will fully assess the robustness and error-handling capabilities of our pipeline. Scalability, cost analysis and stability studies are also required. Privacy concerns must still be addressed. More work is needed to better visualize disease phenotypes with heat maps and dimension-reduced plots. The generalizable of this pipeline to other disease domains such as cardiac, renal, hepatic, and rheumatic diseases should be explored. Limitations in retrieving the correct HPO ID needs to be addressed to assure that term normalization is performed with high accuracy.

Nonetheless, the case for applying LLMs to high-throughput phenotyping is compelling [18], [22], [35]–[37], [58]–[60]. These models are fast, accurate, and ready to run 'out of the box.' Unlike traditional neural network models, they do not rely on an extensive corpus of manual annotations. These models should be generalizable to a variety of diseases without additional training. Current limitations in sign normalization can be addressed using techniques from augmented retrieval generation [40], [61], by additional pre-training, or by creating small specialized models specifically for sign normalization. Large language models such as GPT-4 are expected to become the dominant method for high-throughput clinical text phenotyping.

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
