# OpenReview forum: "High Throughput Phenotyping of Clinical Text Using Large Language Models"
_IEEE.org/EMBS/BHI/2024/Conference — IEEE BHI'24_

### Official Review · Reviewer_Em4g · 2024-08-07
**High-Throughput Phenotyping of Clinical Text Using Large Language Models**

**Overall Rating:** 7
**Confidence:** 3

**Other Quality Metrics:**

(a) Clarity of writing Great;
(b) Clinical Significance Good;
(c) Methodological Novelty Excellent;
(d) Experiments and Results Great;

**Questions For The Authors:**

Can you provide more details on the specific limitations encountered in the sign normalization task and potential strategies to address them?
Given the sensitive nature of clinical data, how do you ensure data privacy and security when using LLMs for phenotyping tasks?

**Strengths:**

The paper uses state of the art llms (GPT-4 and GPT-3.5-Turbo) for high-throughput phenotyping, showcasing the application of advanced NLP techniques in medical informatics.
The application of LLMs highlights their scalability and potential for generalizability across other medical areas by eliminating the requirement for difficult, manually annotated training datasets
The paper provides a thorough evaluation of the models' performance across different tasks, including sign identification, categorization, and normalization, supported by robust experimental results.

**Summary Of The Paper:**

The study "High-Throughput Phenotyping of Clinical Text Using Large Language Models" assesses the use of large language models (LLMs) in automating the phenotyping of clinical summaries from the Online Mendelian Inheritance in Man (OMIM) database. The effectiveness of GPT-4 and GPT-3.5-Turbo in recognizing, classifying, and normalizing clinical indicators is compared in this study. According to the findings, GPT-4 performs better than GPT-3.5-Turbo and achieves good agreement with human annotators. The study shows how LLMs can be used to automate high-throughput phenotyping activities, which can improve generalizability across different phenotyping tasks and decrease the requirement for manually annotated training data.

**Weaknesses:**

The study primarily uses clinical summaries from the OMIM database as a surrogate for real-world physician notes. Validation on actual clinical notes would strengthen the findings and demonstrate real-world applicability.
The paper would benefit from more extensive validation using real-world clinical data to confirm the models' effectiveness in practical settings.
Despite high performance in sign identification and categorization, the models show some limitations in sign normalization

---

### Official Review · Reviewer_mXcy · 2024-08-15
**The paper needs to be revised.**

**Overall Rating:** 5
**Confidence:** 3

**Other Quality Metrics:**

(a) Clarity of writing: good
(b) Clinical Significance: good
(c) Methodological Novelty: fair
(d) Experiments and Results: good

**Questions For The Authors:**

Please compare the results of similar works with your work, and illustrate the novelty of your work compared with the similar ones.

**Strengths:**

The paper proposed a rational pipeline for automating high-throughput phenotypes of clinical text using large language models.

**Summary Of The Paper:**

This study explores the use of large language models (LLMs) to automate the phenotyping of clinical summaries from the Online Mendelian Inheritance in Man (OMIM) database.

**Weaknesses:**

1) More feature reduction methods are needed for comparison.

2) Typos. Such as 'LLMs have a superior ability to extract, summarize, translate, and generate textual information with only a few or even no prompt/fine-tuning samples citeqiu2023large.' and so on.

3) The figures are not visually appealing enough.

---

### Official Review · Reviewer_6bp1 · 2024-08-16
**High-Throughput Phenotyping of Clinical Text Using Large Language Models**

**Overall Rating:** 6
**Confidence:** 4

**Other Quality Metrics:**

Clarity of writing
clinical significance
Novel

**Questions For The Authors:**

1) Which is restriction for the diseases if any?
2) Which is the response time of the system?
3) How can users be involved in the system?

**Strengths:**

The use of a ontology for to support acquisition and processing of vast volumes of unstructured medical related data.
The use of two LLM models.

**Summary Of The Paper:**

This paper proposes a high-throughput phenotyping automates the mapping of patient signs to standardized ontology concepts and
is essential for precision medicine.

**Weaknesses:**

How generic is your method to any medical disease? If no, you should specify these limitations.

---

### Decision · Program_Chairs · 2024-09-23

Accept